# Application of Natural Molecules in Efficient and Stable Perovskite Solar Cells

**DOI:** 10.3390/ma16062163

**Published:** 2023-03-08

**Authors:** Yu Chen, Qian Zhou, Dongmei He, Cong Zhang, Qixin Zhuang, Cheng Gong, Ke Wang, Baibai Liu, Peng He, Yong He, Yuelong Li, Zong-Xiang Xu, Shirong Lu, Pengjun Zhao, Zhigang Zang, Jiangzhao Chen

**Affiliations:** 1Key Laboratory of Optoelectronic Technology & Systems (Ministry of Education), College of Optoelectronic Engineering, Chongqing University, Chongqing 400044, China; 2Institute of Photoelectronic Thin Film Devices and Technology of Nankai University, Key Laboratory of Photoelectronic Thin Film Devices and Technology of Tianjin, Solar Energy Research Center of Nankai University, Tianjin 300350, China; 3Department of Chemistry, Southern University of Science and Technology, Shenzhen 518055, China; 4Department of Material Science and Technology, Taizhou University, Taizhou 318000, China; 5Xinjiang Technical Institute of Physics and Chemistry, Chinese Academy of Sciences, Urumqi 830011, China

**Keywords:** perovskite solar cells, natural molecules, additive engineering, interface engineering, carrier transport layers

## Abstract

Perovskite solar cells (PSCs), one of the most promising photovoltaic technologies, have been widely studied due to their high power conversion efficiency (PCE), low cost, and solution processability. The architecture of PSCs determines that high PCE and stability are highly dependent on each layer and the related interface, where nonradiative recombination occurs. Conventional synthetic chemical materials as modifiers have disadvantages of being toxic and costly. Natural molecules with advantages of low cost, biocompatibility, and being eco-friendly, and have improved PCE and stability by modifying both functional layers and interface. In this review, we discuss the roles of natural molecules on PSCs devices in terms of the perovskite active layer, interface, carrier transport layers (CTLs), and substrate. Finally, the summary and outlook for the future development of natural molecule-modified PSCs are also addressed.

## 1. Introduction

Perovskite solar cells (PSCs) are generating enormous attention in the fields of renewable energy due to their brilliant optoelectronic properties, including high light absorption coefficient, tunable bandgap, and small exciton binding energy [1,2,3,4,5,6,7]. Based on these outstanding optoelectronic properties, scientists have made great advances, and 25.7% power conversion efficiency (PCE) has been achieved in the past ten years [8]. Moreover, the low-cost raw materials and low-temperature solution processability of perovskite has also made it a new star among many photovoltaic technologies, and even among whole renewable energy technology sector.

PSCs originated from dye-sensitized solar cells. Therefore, the architecture is similar to a sandwich, comprising the perovskite layer, charge transport layer and electrode. When the light through the transparent electrode enters the perovskite layer, the electron–hole pairs are generated then separated into free electrons and holes, i.e., photogenerated carriers. The photogenerated carriers will spontaneously diffuse to the corresponding charge transport layer under the concentration gradient, be extracted at the interface between the carrier transport layers (CTLs) and perovskite layer, and finally be collected by the electrode. This process has not been smooth sailing, because there are numerous defects at grain boundaries and the surface in the perovskite film owing to its ionic nature and the rapid crystal growth process. These defects will hinder carrier transport and cause nonradiative recombination. Furthermore, these defects are sensitive to environmental factors, such as moisture, oxygen, temperature, etc., affecting the optoelectronic properties of the perovskite, even leading to degradation. To solve these problems, many scientists have explored various strategies, such as additive engineering, [9,10] interface engineering, [11,12,13], and so on. Various materials with multiple functional groups have been used to assist these strategies. Among these materials, they can mainly divide into two types—synthetic chemical molecules and natural molecules. Synthetic chemical molecules may be toxic, environmentally unfriendly and costly. In contrast, natural molecules have the advantages of low cost, biocompatibility and being eco-friendly (Figure 1). Therefore, natural molecules play a significant role in the field of PSCs.

Recently, many reviews of PSCs have been published and have provided more in-depth insights for researchers [14,15,16,17,18]. However, up to now, few reviews have focused on the development of PSCs modified by natural molecules. It is time to comprehensively summarize recent progress and provide a systematic understanding of natural molecule-modified PSCs. In this review, we first introduce perovskite additive engineering based on natural molecules, and then we discuss the interfacial engineering and CTLs modification and substrate. Finally, we give a conclusion and outlook of natural molecule-modified PSCs.

## 2. Natural Molecule-Based Additive Engineering for PSCs

Perovskite crystals have a soft lattice character and low formation energy; thus, either at grain boundaries or the surface, numerous defects could be formed during the crystallization process [19,20]. Furthermore, unfavorable ion migration and carrier recombination also occurred due to the ionic nature of the perovskite [21,22]. All of this deteriorates the performance of PSCs. To overcome these negative effects, many strategies including composition engineering, dimensionality engineering, nonstoichiometric approach and additive engineering have been intensively developed. Among them, additive engineering using natural molecules with the functional group has been reported to improve the crystallization, passivate the defects, and modify the energetics (Table 1).

1,3,7-trimethylxanthine, which is also named caffeine, is a common natural molecule in people’s daily life. Wang and co-authors added caffeine into perovskite through additive engineering [23]. As is shown in Figure 1a, the conjugated Lewis base with two different chemistry environment carboxyl groups, worked as the “molecular lock” that effectively coordinated with Pb^2+^. As a result, the increased growth activation energy of the perovskite retarded the perovskite crystal growth and induced a preferred orientation. As a result, a champion PCE of 20.25% with an open-circuit voltage of 1.143 V, a short-circuit current (*J*_sc_) of 22.97 mA cm^−2^ and a fill factor of 77.13% has been realized (Figure 1b). Furthermore, the heat stability also improved due to the caffeine can interact with the perovskite again, suppressing the ion migration during the degradation process, leading to the device being stable at 85 °C over 1300 h.

Natural molecules with multiple functional groups provide more interaction with uncoordinated ions. Lin and co-authors employed the M13 bacteriophage as the crystal growth template of the perovskite [24]. As is shown in Figure 1c, the M13 contains four types of amino acids. The lone electron pair on the amino groups of N-terminal alanine and lysine and the negative charge on the carboxyl groups in aspartic acid and glutamic acid on the surface of M13 can effectively be coordinated with Pb^2+^ ions. After adding the M13 into the perovskite, the grains with a diameter from 300 nm to 1 μm are surrounded effectively due to the uniform length of the M13, which led to a homogenous and large crystal size. The M13-added PSC exhibited a champion PCE of 20.1% upon heating the precursor solution at 90 °C compares to the reference PSC with a champion PCE of 17.8%.

The obtained conventional two-step spin-coating lead iodide (PbI_2_) film is too smooth, which is not beneficial for achieving high-quality perovskite films. Liu and co-authors introduced choline chloride (vitamin B4, VB4) into PbI_2_ and constructed a novel heterogeneous PbI_2_ (HLI) structure to change this situation [25]. Through the addition of VB4, the (001) planes of HLI have two interplanar distances of 9.0 Å and 6.9 Å (Figure 1d), which alter the PbI_2_ crystal arrangement, leading to the rough and porous surface of PbI_2_ films that cause the organic precursor to diffuse into films, resulting in better morphology and large grains of the perovskite films. In addition, the negative charge cation vacancy defects of the perovskite films can be effectively passivated by the quaternary ammonium in VB4. With these synergistic effects of VB4, the champion PCE of PSCs is achieved at 22.13% with remarkably elevated *V*_oc_ 1.17 V. They further fabricated PSCs using double-sided modification, and the PCE is up to 24.27% with enhanced *V*_oc_ 1.21 V (Figure 1d).

As we all know, both UV light illumination and oxygen atmosphere extremely accelerate the rate of the degradation process. Lycopene (LCP), the strongest antioxidant, can eliminate both hydrogen peroxide radicals and singlet molecular oxygen produced during photooxidation [26]. Zhuang and co-authors introduced the LCP into the perovskite precursor to achieve efficient and fresh PSCs [27]. As shown in Figure 1e, the LCP molecule is a straight chain containing 11 conjugated double bonds and two non-conjugated double bonds. The long carbon chain makes the perovskite film more hydrophobic, and the LCP could prevent unstable chemical bonds in FA^+^ and MA^+^ from being attacked by oxygen through unsaturated bonds. Moreover, the LCP also can interact with the uncoordinated Pb^2+^ via the Lewis acid–base reaction, which effectively passivates the trap and facilitates a black phase to be formed at room temperature. Through the addition of LCP, the champion PCE was enhanced to 23.62% with a *V*_oc_ of 1.21 V, a *J*_sc_ of 24.10 mA cm^−2^, and an FF of 81.00%. Due to the improved antioxidant ability, 91.7% of the initial average PCE was maintained for 1000 h in oxygen, and 84% was maintained under both UV irradiation and 21% oxygen atmosphere for 8 h compared to the control device of 5 h.

Various defects are generated during the crystallization process. among them, intrinsic site defects and surface defects significantly deteriorate the performance of the PSCs [28]. Guo and co-authors utilized the organic molecule Indigo as an efficient passivator for high-quality perovskite film [29]. The Indigo molecule containing the carbonyl group can interact with the perovskite surface uncoordinated Pb^2+^ and Pb-I antisite defects, and I^#x2212;^ sites can interact with the amino group. Moreover, a hydrogen bond formed between the Indigo and the perovskite surface, which could restrain ion migration. After the treatment of Indigo, the champion PCE of the PSC improved to 23.22%, with a *V*_oc_ of 1.16 V, a *J*_sc_ of 25.02 mA cm^−2^, and an FF of 80%. Furthermore, the Indigo-treated large-area device exhibited a champion PCE of 20.95% with a *J*_sc_ of 24.50 mA cm^−2^, a *V*_oc_ of 1.14 V and an FF of 75%. Moreover, the moisture and thermal stability were enhanced by rich hydrogen bonds and the carbonyl group of Indigo, and 75% of the original PCE of Indigo-treated unencapsulated PSC was maintained after 60 °C aging for 1500 h, and only 15% of the PCE dropped in storage at the ambient environment after the 1500 h aging.

A suitable electronic structure is necessary for perovskite to form a favorable energy match with the adjacent layer [30,31]. Furthermore, the work function (WF) of the underlying layer determines the interface energetics [32]. Therefore, it is imperative to improve the optoelectronic properties of perovskite, including passivate bulk and interface defects, and modify interfacial energetics. Xiong and co-authors employed capsaicin in the perovskite precursor [33]. As is shown in Figure 1f, after the addition of capsaicin, the ultraviolet photoelectron spectra (UPS) exhibited a remarkable decrease in the WF from 4.95 to 4.48 eV, while the Fermi level position (*E*_F_) and the valance band maximum (VBM) increased by 0.48 eV, keeping a constant ionization potential. This means the surface energetics completely transformed from p-type to n-type. Furthermore, Kelvin probe force microscopy (KPFM) showed that a p–n homojunction with 100 nm thickness formed from the perovskite film surface. Therefore, a favorable interface with the electron transport layer (ETL) formed, improving charge transport. Moreover, the capsaicin-containing carbonyl group could effectively interact with the uncoordinated Pb^2+^, leading to both the interface nonradiative recombination and defect-assisted recombination being significantly suppressed. As a result, their p–i–n PSCs achieved a record efficiency of 21.88% with a *V*_oc_ of 1.13 V, a *J*_sc_ of 23.10 mA cm^−2^ and an FF of 83.81%. The capsaicin also improved stability through enhanced hydrophobicity, and the unencapsulated device maintained 90% of initial PCE after storage for 800 h in the ambient environment.

Liu and co-authors introduced vitamin D2 (VD2) into the perovskite bulk [34], and 530 meV of the WF relative to the vacuum energy level (*E*_VAC_) increased, while the *E*_F_ and VBM shifted from 1.09 to 0.51 eV, which indicated that the surface energetics transformed from n-type to p-type. KPFM exhibited that the thickness of the n–p homojunction formed accounts for about 80 nm. Based on the n–p homojunction, the interface barrier between the perovskite and the hole transport layer was minimized (Figure 1g), enhancing the interfacial charge transfer. In addition, the C=C bonds can interact with the uncoordinated Pb^2+^ using the Lewis acid–base reaction, which reduces the defects and improves the crystallization of perovskite. Meanwhile, vitamin C (VC) was introduced into the SnO_2_ ETL, and oxygen defects on the SnO_2_ film surface were reduced, enhancing electron mobility and reducing the interface energy-level offset; thus, the interfacial charge transfer was improved. As a result, this achieved a champion PCE of 24.20% and an FF of 81.01% due to the synergistic effect, and the hysteresis is negligible. Furthermore, 93.04% of the initial PCE of unencapsulated devices was retained after aging 5000 h at room temperature.

The above studies using natural molecules through additive engineering represent important breakthroughs either in crystallization or defect passivation and energetic modification of PSCs, promoting the development of PSCs even in the field of green energy. However, there are still some issues that should be addressed. For example, the PCE still lags behind synthetic chemical additives. In addition, the mechanism between natural molecules and perovskite is explored, while the distribution of natural molecules in perovskite and the interaction with the adjacent layer also should be considered systematically.

**Figure 1 materials-16-02163-f001:**
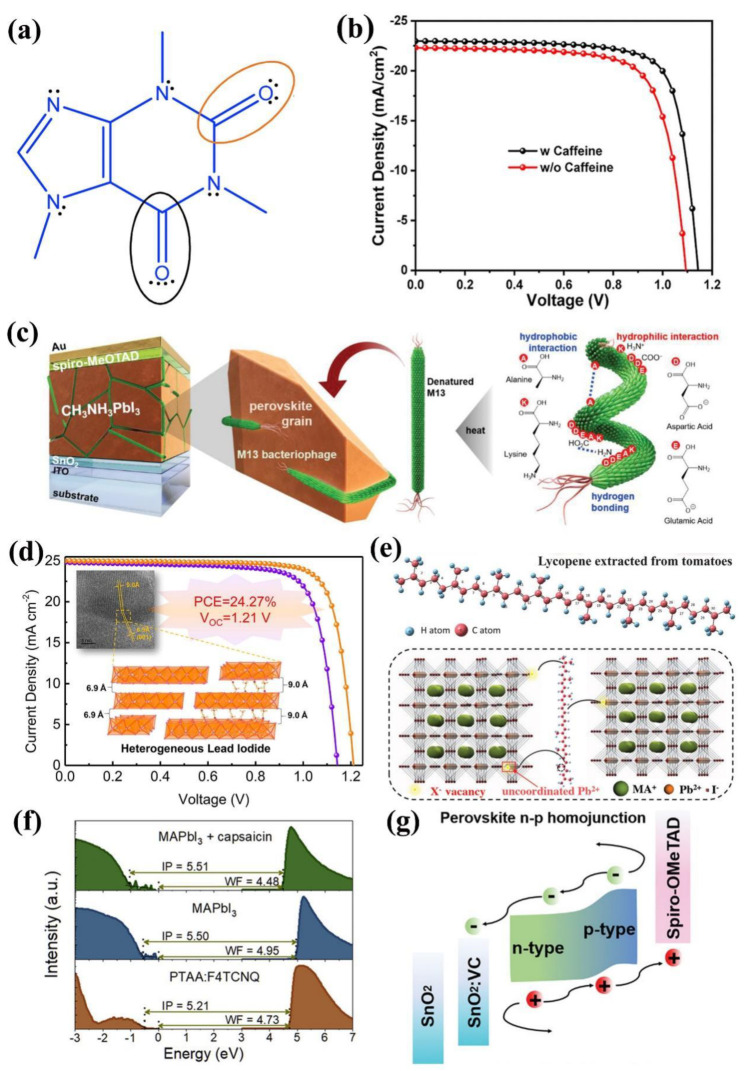
Instances of natural molecule-based additive engineering for PSCs: (**a**) the Lewis chemical structure of caffeine, (**b**) *J-V* curves of the champion PSC of perovskite with and without caffeine in the reverse scan direction [23]. Copyright 2019, Elsevier. (**c**) Illustration of normal-type PSCs structure with the denatured M13 bacteriophage as the perovskite growth template [24]. Copyright 2020, Wiley-VCH. (**d**) *J–V* curves of the champion PSC of perovskite with ordered PbI_2_ films [25]. Copyright 2022, Elsevier. (**e**) The chemical structure of LCP and working as a passivator located at the grain boundary [27]. Copyright 2022, Wiley-VCH. (**f**) UPS spectra of secondary electron cutoff region and valence band region of PTAA: F4TCNQ, perovskite films with and without capsaicin grown on PTAA: F4TCNQ [33]. Copyright 2021, Elsevier. (**g**) Energy-level alignment schematics of the PSC comprising VD2 and VC [34]. Copyright 2022, Wiley-VCH.

## 3. Natural Molecule-Based Interface Engineering for PSCs

The interface is where the carrier extraction, transport and collection occurs, as well as photon transmission. Thus, the properties of the interface are significant for the performance of the PSC, even the stability of the PSC. A promising interface should have these properties, including no energy loss when carriers pass, strong enough to prevent the permeation of oxygen and humidity, as well as ion migration. With this requirement, scientists have put extensive effort into improving the properties of interfaces, including interface contact, interface energetics, and interface trap-states. In this section, some representative studies that have used natural molecules to modify the interface in terms of ETL and HTL are described.

Titanium dioxide (TiO_2_) is a commonly used ETL in PSCs, due to its advantages of high stability and excellent optoelectronic properties [35,36]. However, the ultraviolet photocatalysis effect and tremendous oxygen vacancies on the TiO_2_ surface can cause degradation of the perovskite. In addition, the energy level of TiO_2_ and perovskite are not well matched. All the above leads to poor performance of PSCs. To ameliorate the situation, You and co-authors used the biopolymer heparin sodium (HS) with multiple functional groups as the interlayer anchoring the TiO_2_ and perovskite [37]. After treatment with HS, interaction of the ionic functional groups, including -COO^−^, SO_3_^−^ anionic in HS with uncoordinated Pb^2+^ in perovskite and Ti^4+^ in TiO_2_ occurred, with the Na^+^ cationic group prone to filling the vacancies of MA^+^ and interacting with the uncoordinated I^−^. All led to ameliorated morphology of the TiO_2_ surface with better hydrophilicity and no pinhole, and provided a homogeneous surface for the growth of perovskite, resulting in the perovskite film with fewer surface defects and better crystallinity. Furthermore, the strong anchoring effect of HS effectively hinders ion migration and suppresses hysteresis. As a result, this realized a champion PCE of 20.1% and improved long-term stability of 90% and 85% original PCE, which was maintained after 70 d storage at nitrogen and air, respectively.

Zhang and co-authors introduced dopamine (DA) to passivate the TiO_2_ surface using a chelating effect in n–i–p planar PSCs [38]. As is shown in Figure 2a, after the modification of DA, the enediol ligands of DA formed a conjugated structure with the surface Ti atoms, leading to a compact and dense TiO_2_ film, effectively suppressing deep trap-states and reducing oxygen vacancies. Moreover, the uncoordinated Pb^2+^ and the Pb-I/Br antisite defects on the surface of perovskite can be passivated by the terminal amino groups in DA. Therefore, the DA-modified TiO_2_ can be used as a crosslinker in the interface, effectively accelerating charge transfer and reducing charge accumulation at the surface between TiO_2_ and perovskite. As the result, this realized a champion PCE of 20.93% with negligible hysteresis, and 80% of the initial PCE of the unencapsulated device was maintained after 1200 h of continued full-sun illumination without any UV filter.

As for alternative ETL materials, SnO_2_ has the advantages of low-temperature manufacturing and good energy-level alignment. However, the inevitable surface defects and pinholes of the film significantly deteriorate the photovoltaic performance of PSCs. To overcome these shortages, Geng and co-authors employed L-aspartic acid (LAA) as the interlayer between ETL and perovskite [39]. As shown in Figure 2b, with the treatment of LAA, the carboxyl group interacts with uncoordinated Sn^4+^ and neutralizes the alkalinity of the hydroxyl group in SnO_2_, leading to a denser and smoother ETL layer with reduced defects, and negligible transmittance loss, and enhanced conductivity. Moreover, hydrogen bonds could be formed between the amino group of LAA and halogen ions of the perovskite, resulting in improved crystallization and inhibited ion migration. Thus, the ETL layer and the perovskite layer are connected chemically via LAA. In addition, the conduction band (ECB) of LAA-modified SnO_2_ decreased from −4.17 eV to −3.96 eV, which is more aligned with the perovskite layer, facilitating the charge transfer. As a result, a champion PCE of 22.63% with a *V*_oc_ of 1.16 V, *J*_sc_ of 24.75 mA cm^−2^ and an FF of 79.32% was achieved, and the PCE only declined by 12.8% after aging 2184 h at 60 °C and 9.1% declined after aging 1680 h at a relative humidity of 40%.

The phenomenon of charge transport imbalance and charge accumulation at the interface result from the discrepant extraction rates, which is a bottleneck phenomenon of the PSC, and influences device performance [40]. To overcome this challenge, Kim and co-authors introduced 2-[carbamimidoyl(methyl)amino]acetic acid (creatine) into the interface between the SnO_2_ ETL and perovskite [41]. Creatine possesses high polarity and functional groups including the carbamimidoyl group and the carboxyl group. Amines and ammonium ions or an intermediate state between them could be converted from the carbamimidoyl group through a resonance structure. With the treatment of creatine, the WF of SnO_2_ decreased from 3.6 eV to 3.32 eV, and the conductivity increased by 0.44 × 10^−3^ mS cm^−1^. Furthermore, superfluous amines and ammonium ions can interact with halides and uncoordinated Pb^2+^ in the perovskite through electrostatic interaction, which facilitates charge transfer and prevents back-recombination. Furthermore, with the interlayer of creatine, the Urbach energy of the perovskite layer is slightly decreased, implying that a low level of energetic disorder resulted in a low defect density, and creatine passivates the defect states, improving the quality of the perovskite layer. As a result, a champion PCE of 20.8% with a *V*_oc_ of 1.19 V, *J*_sc_ of 23.4 mA cm^−2^ and an FF of 75.9% was realized. The device with creatine exhibited better stability than the control, with over 90% of the original PCE of unencapsulated devices retained after storage for 50 days under ambient conditions.

PSCs with solely SnO_2_ ETL without any modification always exhibited poor electron extraction due to the appeared trap-states. To cover this shortage, 3-amino-4-pyrazolecarboxylic acid (APA) was employed by Chen and co-authors as the interlayer between the ETL and perovskite for ameliorating the optoelectronic properties of the buried perovskite film interface and ETL [42]. With the introduction of APA, the carboxyl group with lone pairs of electrons in APA could effectively coordinate with the uncoordinated Sn^2+^, leading to trap-states being reduced from 8.28 × 10^16^ cm^−3^ to 6.73 × 10^16^ cm^−3^, and electron mobility and conductivity are also improved. In addition, the difference between *E*_F_ and VBM decreased by 0.12 eV, which means the APA-treated SnO_2_ is more n-type, facilitating charge transfer and collection. Moreover, carboxyl and pyrazole N are electron-rich units that could interact with uncoordinated Pb^2+^ through coordination and hydrogen bonding, which improves crystallization and regulates crystal growth. Therefore, APA works as a multifunctional bridge, chemically linked to the SnO_2_ and the buried perovskite interface, leading to an impressive champion PCE of 24.71% with a *V*_oc_ of 1.159 V, *J*_sc_ of 25.51 mA cm^−2^ and an FF of 83.56%. In addition, over 83% of the original PCE was retained after aging in nitrogen for 2400 h at room temperature in the dark for the unencapsulated devices, and over 80% of the initial PCE was retained under 1 sun illumination of continuous maximum output power point (MPP) tracking after 500 h.

In inverted-structure (p–i–n) PSCs, nickel oxide (NiO_X_) is a promised HTL material due to the superiorities of high hole transport ability and excellent stability. However, imperfect film coverage and pinholes commonly arise during the solution fabrication process [43,44,45]. Meanwhile, the energy level of NiO_X_ film is mismatched with the perovskite, leading to a decrease in carrier transport ability. To overcome these disadvantages, Xie and co-authors used adenine as the interlayer between NiO_X_ and perovskite to regulate the energy level in PSCs [46]. Adenine is a kind of nucleobase that contains the easy-to-take redox reaction under low oxidation potential DNA and RNA polymers. With the modification of adenine, the WF and VBM dropped to 4.54 eV and 0.86 eV, respectively, leading to the highest occupied molecular orbital (HOMO) level shifted to 5.4 eV (Figure 2c), minimizing the energetic mismatch of HTL and perovskite, and increased the *V*_oc_. Furthermore, the Ni^3+^/Ni^2+^ ratio increased from 2.96 to 3.45, resulting in better hole conductivity. Moreover, the three main peaks of perovskite deposited on the NiO_X_/adenine were enhanced, indicating improved crystallization. Thus, the adenine worker as the surface modifier ameliorates the energy-level mismatch, improves the crystallization of perovskite, and enhances charge transport and extraction. As a result, a champion PCE of 18.96% with a *V*_oc_ of 1.06, V, *J*_sc_ of 22.94 mA cm^−2^, and an FF of 77.76% was realized, and 90% of the initial PCE of the adenine-modified devices without encapsulation maintained after 600 h storage at room temperature under ambient conditions (50–60% relative humidity).

One of the Fullerene derivatives, [6,6]-phenyl-C61-butyric acid methyl ester (PCBM), in inverted-structure PSCs is widely used ETL material due to its additional function of passivating perovskite film surface defects and charge traps [47,48]. However, the problem of energy-level mismatch between the PCBM and metal electrode severely impedes electron extraction. To solve this problem, Xiong and co-authors inserted the natural molecules Isatin and Isatin-Cl between ETL and the aluminum (Al) cathode electrode [49]. After introducing the Isatin and Isatin-Cl, the WF of the Al electrode significantly decreased by 0.63 eV and 0.87 eV, respectively, which is attributed to the interfacial negative dipole formed between the ETL and electrode, leading to the lowest unoccupied molecular orbital (LUMO) of PCBM, higher than the WF of Al electrode, facilitating electron transport. As a result, the device with Isatin-Cl realized a champion PCE of 19.74% with a *V*_oc_ of 1.086 V, *J*_sc_ of 22.27 mA cm^−2^ and an FF of 81.63%, while the device with Isatin had 19.25% with a *V*_oc_ of 1.081 V, *J*_sc_ of 22.22 mA cm^−2^ and an FF of 80.15%.

These representative works show the application of natural molecules as interface engineering materials, promoting the development of the application of natural molecules in PSCs, and bring PSCs closer to commercial applications. Although the thermodynamics and dynamics of the carriers’ extraction, transport, and collection must be considered, the stability of the natural molecule before and after reacting with adjacent layers may be ignored, which may affect device performance and stability. Last, the cost of an extra interlayer should also be mentioned, which may cause a sharp rise in costs in commercial application.

**Figure 2 materials-16-02163-f002:**
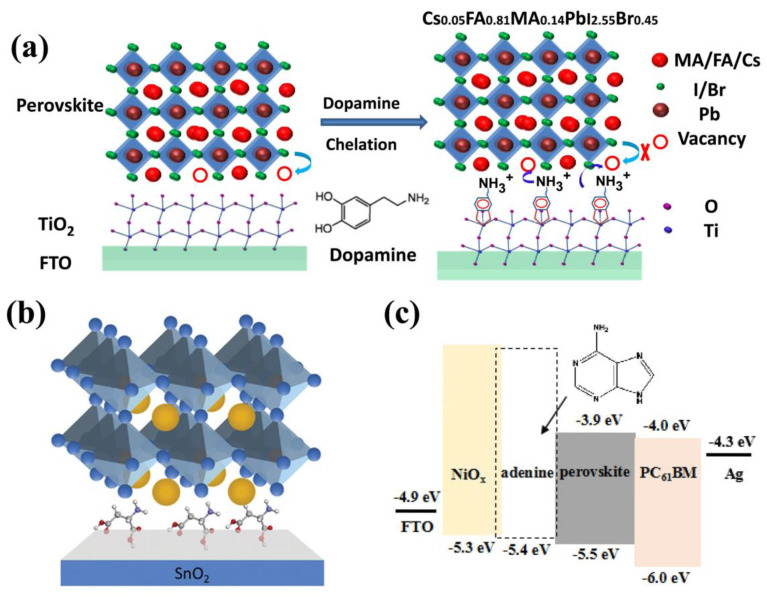
Instances of natural molecule-based interface engineering for PSCs: (**a**) Schematic diagram of dopamine function between the perovskite and TiO_2_ interface [38]. Copyright 2019, Elsevier. (**b**) Passivation mechanism of LAA at the SnO_2_ ETL/PVK interface [39]. Copyright 2020, Wiley-VCH. (**c**) The energy diagram of PSCs and molecule structure of adenine [46]. Copyright 2020, Elsevier.

## 4. Natural Molecule-Modified Carrier Transport Layers

Both ETL and HTL are the CTLs. They separate and extract photogenerated carriers, while blocking counter carriers [50,51]. In addition, the overlying layer is directly affected by the chemical and surface properties of the CTL, such as the perovskite film, which further influences charge separation, transportation, and the final performance of the device. Thus, developing a high-quality CTL is significant for realizing high-performance PSCs. An ideal CTL should have high transmittance, low trap-state, suitable band alignment with adjacent layers, and proper conductivity that diminishes the interface charge accumulation and recombination. The commonly used CTL, including ETL (TiO_2_, SnO_2_, PCBM) and HTL (PEDOT: DSS, PTAA, NiO_X_) are suffering from their own disadvantages. To overcome these shortages, scientists have made efforts to improve the properties of the CTL, including dopants and substitutes (Table 1). Therefore, in this section, some representative work that uses natural molecules to improve the properties of HTL and ETL are described.

**Table 1 materials-16-02163-t001:** Summary of natural molecules applied in PSCs.

Year	Natural Molecules	Methodology	PCE	Ref.
2019	Caffeine	Additive engineering	19.8%	[23]
2020	M13	Additive engineering	20.1%	[24]
2021	Capsaicin	Additive engineering	21.88%	[33]
2022	L-Theanine	Additive engineering	24.58%	[52]
2022	Indigo	Additive engineering	23.22%	[29]
2022	Vitamin B4	Additive engineering	24.27%	[25]
2018	Heparin Sodium	Interface engineering	20.1%	[37]
2019	Dopamine	Interface engineering	20.93%	[38]
2020	Creatine	Interface engineering	22.1%	[41]
2020	Adenine	Interface engineering	18.96%	[46]
2023	3-amino-4-pyrazolecarboxylic acid	Interface engineering	24.71%	[42]
2019	Isatin and Isatin-Cl	Interface engineering	19.74%	[49]
2018	Dopamine	CTLs modification	18.5%	[53]
2017	Dopamine	CTLs modification	16.6%	[54]
2018	Cellulose paper	Substrate	9.5%	[55]
2019	Bamboo-cellulose fibrils	Substrate	11.68%	[56]

TiO_2_ is a commonly used ETL material, but the electron conductivity, trap-state and mismatched band alignment level are still the challenges for high-performance PSCs. To change this situation, Peng and co-authors used DNA as a dopant into *meso*-TiO_2_ using the hydrothermal method [57]. After being doped by DNA, a positive charge in TiO_2_ particles could strongly interact with DNA-containing negative charges, therefore surrounding them with DNA, improving the crystallization and morphology of TiO_2_ film, and the high-quality *meso*-TiO_2_ film with better conductivity, increased hydrophilicity, and reduced trap-state. Furthermore, the *E*_F_ of DNA-doped *meso*-TiO_2_ film is increased by 0.1 eV (Figure 3A), so the energy barrier between the perovskite and ETL is decreased. As a result, the device with DNA-doped *meso*-TiO_2_ exhibited a higher PCE of 17.59% with a *V*_oc_ of 1.068 V, *J*_sc_ of 22.90 mA cm^−2^ and an FF of 71.90%.

Apart from the improvement of the PCE of PSCs, stability and potential Pb leakage should also be concerned. Mokhtar and co-authors indicated a bioinspired underlying solution to Pb isolation using hydroxyapatite (HAP) [58]. With the mixture of HAP nanoparticles, the two nanoparticles formed a scaffold for the PSCs. The HAP in the blended scaffold exhibited an average pore size of 36, 52, and 62 nm corresponding to 0%, 30%, and 70% HAP/TiO_2_ ratio, respectively. Therefore, there is more space for perovskite grain growth. Furthermore, the HAP is dispersed vertically within the percolated TiO_2_ nanoparticle network, and the phosphate (PO_4_^3−^) group in HAP can interact with Pb^2+^, benefiting charge extraction and transport. As a result, this achieved a champion PCE of 20.98% with a *V*_oc_ of 1.076 V, *J*_sc_ of 24.73 mA cm^−2^ and an FF of 78.85%, and similar stability of 0% to 70% HAP with over 85% of their initial PCEs maintained after storage under ambient conditions without encapsulation. Moreover, HAP exhibited excellent Pb^2+^ absorption capacity with 1350 mg g^−1^; the one-point broken devices with 70% HAP scaffold and 20 or 40 μm HAP encapsulation layer released Pb concentration in water are 0.75 and 0.55 ppm, respectively, lower than the 0% and HAP-free devices of 7.0 ppm (Figure 3B). Thus, HAP not only improves performance of PSCs but also impedes Pb leakage.

Compared to TiO_2_ ETL, SnO_2_ as ETL has advantages of simple processing, suitable energy-level alignment, and high carrier mobility [59,60]. However, an SnO_2_ ETL suffers from lattice mismatch with perovskite, which leads to interfacial stress and defects. In addition, oxygen vacancies are also generated during the low-temperature manufacturing process, which induces nonradiative recombination [61]. To solve these problems, Yu and co-authors treated SnO_2_ with tyrosine (Tyr) to improve the properties of SnO_2_ film, including passivating the interface defects and tuning the energy level [62]. After the SnO_2_ film was doped with Tyr, the oxygen vacancies were filled by Tyr through the interaction between the carboxyl group and Sn^4+^. Furthermore, the conduction band minimum (CBM) of Tyr-modified SnO_2_ increased from the pristine −4.34 eV to −4.15 eV, which diminishes the energy loss and promotes electron extraction. Moreover, the amino group in Tyr on the surface of the SnO_2_ film could interact with the I ion, and work as the bridge between perovskite and ETL, which not only eliminates the gap between the two layers facilitating charge transfer at the interface, but also improves the crystallization to render less grain boundary and preferable crystal orientation. As a result, the Tyr-modified device achieved a champion PCE of 22.17% with a *V*_oc_ of 1.11 V, *J*_sc_ of 24.95 mA cm^−2^ and an FF of 80.05%, and 87% of the original PCE of the unencapsulated device was maintained after aging 864 h in ambient air (25 °C, 25% ± 5% relative humidity).

Although commercial SnO_2_ colloid solution was widely used for ETL preparation, the soft agglomerate phenomenon of the SnO_2_ colloid solution often appeared when under long-term storage due to van der Waals attraction, leading to incompletely covered film [63,64]. To overcome this shortage, Wang and co-authors employed two amino acids, glycine (GLY) and alanine (CA), and ammonium titanium fluoride (ATF) to ameliorate this situation [65]. As shown in Figure 3C, the fresh SnO_2_ exhibited its main peak at 9 nm, while the agglomerated SnO_2_ nanoparticle corresponded to 100 nm. In contrast, there is only a single peak at 10 nm after the addition of CLY or CA, which indicates that amino acids could effectively prevent soft agglomeration. When the ATF was added to the SnO_2_ colloid solution, the average size increased to 1.4 μm due to the fluoride ion decreasing the Zeta potential. After 30 d aging, the control and GLY SnO_2_ colloid solution observed two peaks. When the CA SnO_2_ colloid solution retained a single peak, this indicated that it was more stable for long-term storage. Furthermore, the carboxyl group interacted with the hydroxyl group on the surface of SnO_2_, and the carbonyl group passivated the oxygen vacancies via coordination with Sn^4+^, leading to an ameliorated morphology of SnO_2_ film. Meanwhile, the negatively charged defects of perovskite are effectively passivated by the amino group in GLY and CA, resulting in an improvement in crystallization. As a result, the PSC modified by GLY exhibited a champion PCE of 19.71%, with a *V*_oc_ of 1.18 V, a *J*_sc_ of 22.83 mA cm^−2^ and an FF of 73%, and the encapsulated device maintained 83.58% of the original PCE after exposure to an 85% relative humidity environment for 360 h.

As the common HTL material in inverted PSCs, PEDOT:PSS has intrinsic properties including high conductivity, favorable transparency, and low-temperature solution processing [66,67,68,69]. However, there are still some disadvantages that should be considered, such as high acidity and low WF (5.1 eV). To overcome its shortages, Huang and co-authors obtained the copolymer DA-PEDOT:PSS by adding ammonium persulfate into the mixture of DA, PEDOT and PSS. With the doped DA, the WF of copolymer DA-PEDOT:PSS decreased by 0.23 eV, all the better for the alignment with the valence band (VB) of −5.4 eV, reducing the energy barrier and benefit charge transfer. Furthermore, the pH value of copolymer DA-PEDOT:PSS is 5.2, while PEDOT:PSS is 1.8. The decreased acidity means that the DA-PEDOT:PSS would generate less acid corrosion. Thus, the champion PCE improved to 16.6% better than the control of 15.2%. The unencapsulated DA-PEDOT:PSS-based device retained 85.4% of the initial PCE after aging 28 d under nitrogen conditions. Although the DA has improved the properties of PEDOT:PSS through copolymerization, the potential effect of DA on the HTL should be explored in depth. To clear these questions, Xue and co-authors investigated the effects of DA-doped PEDOT:PSS systematically [53]. As shown in Figure 3D, the DA-PEDOT:PSS film contact angle increased after annealing, which was attributed to the crosslinking reaction that occurred between DA and PEDOT. The surface with higher hydrophobicity of HTL is beneficial for the crystallization of perovskite and enhances waterproofness. Furthermore, the uncoordinated Pb^2+^ on the buried surface of the perovskite can be passivated by the amino and hydroxyl groups (Figure 3E), which effectively passivated surface defects and suppressed nonradiative recombination, promoting charge extraction. Moreover, the DA-PEDOT:PSS matched with the VB of perovskite well, due to deeper WF (5.33 eV), thus maximizing the built-in potential. As a result, a champion PCE of 18.5% with an increased *V*_oc_ of 1.08 V was achieved. This exploration provides an essential guideline for designing a new HTL material of inverted PSCs with ameliorated performance.

For the improvement of the properties of HTLs, besides the modification approach, developing new substitute materials is another way. Yusoff and co-authors employed DNA–hexadecyl trimethyl ammonium chloride (CTMA) as the HTL in inverted PSCs [70]. Figure 3F shows that the HOMO and LUMO of DNA-CTMA are estimated at 5.4 eV and 1.1 eV, respectively. The HOMO energy level of DNA-CTMA is more matched with the VB of perovskite (−5.4 eV), facilitating hole extraction, and the higher LUMO level of DNA-CTMA could block electrons effectively. Furthermore, DNA-CTMA film exhibited excellent transparency (300–1100 nm) convenient for the light to pass through. Furthermore, an approximate 6% initial weight loss (the bound water) below 140 °C, which kept stable when temperature reached 225 °C, confirmed that DNA-CTMA has good thermal stability. As a result, a champion PCE of 15.86% with *V*_OC_ of 1.04 V, *J*_SC_ of 20.85 mA cm^−2^ and an FF of 73.15% was realized for the device with DNA-CTMA HTL.

CTLs are an important part of PSCs, and these works represent great progress in the use of natural molecule-modified CTLs in PSCs. However, most are concentrated on one question of CTLs, and cannot use a natural molecule to achieve defect passivation, energetics modification, and suppression migration simultaneously. Thus, there is still a long way to go for promising CTLs.

**Figure 3 materials-16-02163-f003:**
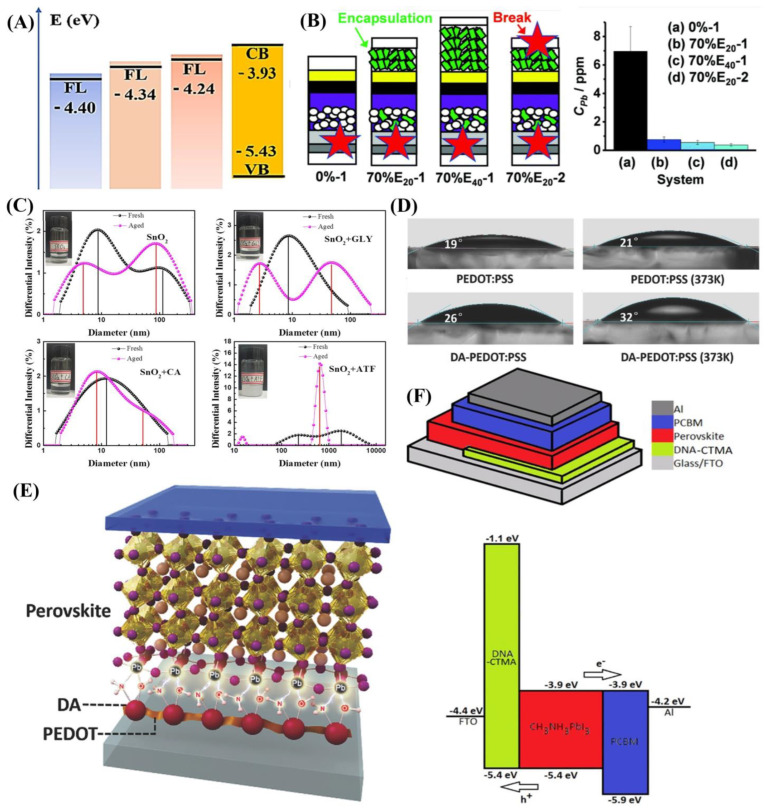
Instances of natural molecule-modified CTLs. (**A**) Schematic illustration of the energy diagram [57]. Copyright 2020, Elsevier. (**B**) The breakage point in devices and Pb release after 24 h in water [58]. Copyright 2021, Royal Society of Chemistry. (**C**) DLS spectra of fresh and aged SnO_2_ solution with different additives [65]. Copyright 2021, Elsevier. (**D**) Temperature-dependent contact angles of PEDOT: PSS and DA-PEDOT: PSS. (**E**) Potential passivation effect [53]. Copyright 2018, Wiley-VCH. (**F**) Device structure and energy band alignment of DNA-CTMA-based PSCs [70]. Copyright 2016, Wiley-VCH.

## 5. Natural Molecule-Based Flexible Substrate

Flexible PSCs are commonly coated on plastic substrates including polyimide (PI), polyethylene terephthalate (PET), and polyethylene naphthalate (PEN) due to their advantages of excellent mechanical and chemical stability [71,72,73]. However, the extremely long time for degradation or decomposition in nature results in environmental pollution. Therefore, it is imperative to develop alternative biocompatible substrates for flexible PSCs.

In 2018, Gao and co-authors first employed biocompatible cellulose paper with low cost as the substrate of flexible PSCs [55]. The substrate modified by carbon not only exhibited good conductivity of 14.2 Ω sq^−1^ and a smoother surface, but also showed an aligned energy level with perovskite (Figure 4a). As a result, a champion PCE of 9.05% was achieved for the bio-substrate device without HTL, and good bending stability with 75% initial PCE was maintained after 1000 bending cycles (Figure 4b). Later, Zhu and co-authors reported a bamboo-cellulose nanofibril (b-CNF) substrate for flexible PSCs [56]. The b-CNF substrate exhibited brilliant transparency, and over 90% visible light (400–800 nm) could transmit through the paper. Combined with 150 nm indium zinc oxide (IZO), ultrahigh flexibility and good transmittance b-CNF/IZO electrodes with high conductivity were obtained. Moreover, the b-CNF/IZO electrode showed good mechanical stability with Young’s modulus of 7.3 GPa and tensile strength of 230 MPa (Figure 4c). In addition, the electrode remains a stable square resistance of 40 Ω sq^−1^ after 3000 bending tests at a 4 nm curvature radius, while the PEN/IZO electrode increased after 2400 bending tests (Figure 4d). As a result, the device fabricated on this substrate realized a champion PCE of 11.68% with *V*_OC_ of 0.935 V, *J*_SC_ of 16.92 mA cm^−2^ and an FF of 73.86%, and retained 70% of initial PCE after bending 1000 times at a 4 mm curvature radius.

These studies represent great progress in flexible substrates, which means that environmentally unfriendly plastic substrates could be substituted by the biocompatible flexible substrate, and provide a new direction for flexible PSCs. However, there are still some problems to be solved. For example, although a biodegradable flexible substrate shows good flexibility and light transmittance, its device performance still needs to be greatly improved. In addition, the environmental stability of this type of substrate also should be addressed.

## 6. Summary and Outlook

This review concentrated on the recent progress of PSCs modified by natural molecules with high performance and stability. Compared to synthetic chemical molecules, Natural molecules are biocompatible and eco-friendly. More important, natural molecules play an important role either in perovskite or interface and CTL. Based on additive engineering, the introduced natural molecules improve the crystallization process and passivate the defects as well as the energetics modification. Through interface engineering, the employed natural molecules ameliorate interface contact and energy-level alignment, promoting charge transfer. Moreover, CTL doped by natural molecules improved the surface and chemical properties that benefit the overlying layer deposition, and enhanced charge separation and transportation. For the device substrate, natural molecule-based substrates exhibited brilliant flexible ability, excellent transmittance, and biocompatibility. In summary, natural molecules exhibited valuable functions in PSCs, and have a broader application potential. However, there are still some issues that should be considered.

First, the PCE of natural molecule-modified PSCs lags behind the PSCs modified by synthetic chemical materials. Thus, it is necessary to develop some natural molecules that can effectively promote the PCE of PSCs.

Second, when perovskite is modified with additive engineering, the distribution of natural molecules in it and the interaction with adjacent layers also need to be noted. The mechanism of natural molecules with perovskite and adjacent layers also should be investigated systematically.

Third, inspired by the natural molecule-based substrate, the CTL could also be substituted for natural molecules.

## Data Availability

Not applicable.

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
