# Peer review of "Application of Natural Molecules in Efficient and Stable Perovskite Solar Cells"

_materials, 2023, doi:10.3390/ma16062163_

Round 1

Reviewer 1 Report

Review report of Application of natural molecules in efficient and stable perovskite solar cells.

This review concentrated on the recent progress of Perovskites Solar Cells (PSCs)modified by natural molecules with high performance and stability.
I find that this review paper has some value in that it has pulled together a list of references in roles of natural molecules on PSCs in terms of the perovskite as an active layer, interface, carrier transport layer, and substrate. It has enumerated various synthesis techniques and optimization procedures that researchers have chosen to work primarily with in their publications.  A review paper would normally provide a critical review of many of the papers from a variety of groups.  In addition, I find too much emphasis on cataloging the approaches of each paper rather than providing an effective analysis of the literature that provides guidance to the reader as to which works represent the key results that everyone needs to be aware of. 

I find that the paper would benefit greatly if it were to delve more deeply into a few of the novel approaches, explain what it is that those approaches are trying to accomplish that is different than conventional approaches, describe more of the physical mechanisms involved in those approaches, explain more about the trade-offs in design parameters involved in the novel stabilized high efficient approaches, and give the authors' perspectives on what future directions and applications will be most beneficial for Perovskites solar cells.  Indeed, this last comment is key to revising the work to be more effective.  With this keynote, it’s the area highly recommended to publish new approaches-based reviews. My reading suggests that this work is well carried out and well-presented but also that the manuscript does not contain sufficient scientific novelty to attract the readership in terms of deep analysis.

Reviewer 2 Report

The manuscript presents a review regarding the natural molecules on Perovskite Solar Cells that enhance the device performance in terms of the perovskite active layer, interface, carrier transport and substrate, which is interesting to be discussed. However, after reading the article carefully, we have concluded that the article still needs some changes before it is ready for publication.

1.           First and foremost, the article submitted is not thoroughly discussed and cannot be considered as a comprehensive review article. A complimentary review paper must present a systematic review of literature on recent and key articles in the research field. It is then completed by identifying research gaps and the author is expected to recommend new area of research. The latter part is the most crucial element in a review article.

2.           In general, a table listing related articles that have been previously reported is indispensable, showing a comparison of all key parameters such as year, methodology and most importantly output results.

3.           The conclusion section should state the implications of the findings and an identifies possible new research fields.

4.           It is suggested to also conclude the review article with a figure, showing the evolution of natural molecular development in PSC over the years. It may help in providing a clear overview of the use of natural molecules in PSC in the future.

Reviewer 3 Report

The group of authors has described the possible engineering process to obtain comparatively stable, robust, and highly efficient perovskite solar cells. The topic is very interesting in terms of the modern days sustainable energy generation for a future green globe. However, the manuscript required a comprehensive revision before publishing.

·         There are some anomalous statements including many eye-catching typos in the introduction, so it is highly recommended to rewrite the introduction.

·         Also, the introduction is lacking the state-of-the-art of the perovskite solar cells, missing some recent references (research articles and review articles), such as doi.org/10.1002/cnma.202200471, and more similar ones.

·         In section 2, lines 79-80, “Perovskite is drinking coffee” is very interesting. Wang and co-workers added 1,3,7- 79 tri methylxanthine (caffeine) into the perovskite.[22]……..Doesn’t make any comprehensive outlook by using someone's dialogue for such kind of article.

·         The word “co-workers” should be changed to either “co-authors” or “ First Author name et al., or the research group”.

·         Some key abbreviations are missing.

·         Figures 3, and 4, are partially questionable. Some areas such as Figures 3g and 4b quality must be improved.

·         As per the title and abstract, the conclusion is still incomplete. Some key issues are mentioned but there should be some possible directions for further addressing those issues to achieve highly stable and super-performing PSCs.
